# Búsqueda Tabú Multi-Entorno para el Problema Colaborativo de Rutas de Vehículos con Clientes Compartidos

**Juan Aday Siverio-González**
Departamento de Ingeniería Informática y de Sistemas
Instituto Universitario de Desarrollo Regional
Universidad de La Laguna
alu0101503950@ull.edu.es

**Belén Melián-Batista**
Departamento de Ingeniería Informática y de Sistemas
Instituto Universitario de Desarrollo Regional
Universidad de La Laguna
mbmelian@ull.edu.es

**J. Marcos Moreno-Vega**
Departamento de Ingeniería Informática y de Sistemas
Instituto Universitario de Desarrollo Regional
Universidad de La Laguna
jmmoreno@ull.edu.es

## Abstract

El problema de rutas de vehículos con clientes compartidos es un problema de optimización combinatoria que surge en la logística urbana, donde múltiples transportistas operan de manera independiente, pero comparten un subconjunto de clientes. La resolución eficiente de este problema puede generar importantes ahorros en los costes operativos y mejorar la utilización de los recursos en la distribución de la última milla. Este trabajo presenta un enfoque metaheurístico basado en Búsqueda Tabú para resolver el problema considerado. El algoritmo propuesto incorpora multi-entornos y hace uso de técnicas de oscilación aleatoria para explorar eficientemente el espacio de soluciones. Los experimentos computacionales realizados corroboran la eficacia de la Búsqueda Tabú frente al enfoque más reciente de la literatura, que combina un procedimiento GRASP con búsqueda local iterativa. La Búsqueda Tabú supera al estado del arte, encontrando soluciones de mayor calidad en tiempos computacionales significativamente más cortos, logrando reducciones de costos notables. Los resultados resaltan el potencial de la Búsqueda Tabú para resolver problemas complejos de rutas de vehículos en marcos colaborativos, lo que la convierte en una alternativa prometedora para aplicaciones logísticas del mundo real.

## 1. Introducción

El transporte internacional de mercancías juega un papel relevante en el desarrollo y consolidación de importantes sectores económicos como el comercio, la industria o el turismo. En la distribución de mercancías intervienen numerosos agentes y medios de transporte integrados en complejas cadenas

XVI XVI Congreso Español de Metaheurísticas, Algoritmos Evolutivos y Bioinspirados (maeb 2025).

de suministro constituidas por centros de producción, almacenes de consolidación y distribución, redes de transporte terrestre, puertos, aeropuertos, transportistas y clientes finales.

Para el reparto de la última milla se emplean las infraestructuras y redes de transporte usadas por el resto de actividades socioeconómicas relacionadas con la movilidad terrestre. La presencia simultánea de vehículos de reparto de mercancías, vehículos de transporte público de personas y vehículos particulares congestiona las vías de circulación con el consiguiente incremento de los tiempos de desplazamiento y de las emisiones de gases de efecto invernadero. Según el Ministerio de Agricultura y Pesca, Alimentación y Medioambiente del Gobierno de España, el transporte de mercancías y personas es responsable del $30,7\%$ de las emisiones totales de gases de efecto invernadero, representando el transporte por carretera por sí solo el $28,4\%$ del total de emisiones. Sin embargo, no es este el único efecto no deseado del uso intensivo que hacemos de los vehículos de transporte terrestre. En las grandes ciudades, la congestión del tráfico está detrás de la pérdida de calidad de vida de buena parte de la población y de la disminución de la competitividad empresarial.

Entre las medidas que pueden implementarse para reducir estos efectos destacan las que abogan por una gestión inteligente del reparto de mercancías a través del transporte colaborativo. Los modelos de transporte colaborativos se posicionan como alternativas adecuadas para el reparto de mercancías. Suponen un cambio de paradigma al fomentar el uso compartido de información, clientes, redes, instalaciones o vehículos por parte de los diferentes actores del transporte. Existen tres tipos de estructuras de colaboración. En la colaboración vertical, esta se produce entre actores de la misma cadena de suministros, aunque estos deben estar situados en distintos niveles dentro de la cadena (por ejemplo, colaboración entre importadores y transportistas). En la colaboración horizontal, dos o más actores no relacionados, y que habitualmente compiten en el mismo mercado, deciden compartir información, clientes o capacidades para operar colaborativamente (por ejemplo, diseñando rutas de entrega que eviten múltiples visitas a un mismo cliente). En la colaboración lateral se produce una combinación de colaboración vertical y horizontal.

Además de sus evidentes ventajas sociales y medioambientales, la apuesta por el transporte colaborativo hace que las empresas aumenten sus beneficios, reduzcan sus costes y mejoren el grado de satisfacción de sus clientes.

Dentro de la colaboración horizontal destacan dos procedimientos operacionales: compartir pedido (*order sharing*) y compartir capacidad (*capacity sharing*). Desde el punto de vista logístico, son de relevancia los sistemas colaborativos que involucran la gestión eficiente de los clientes compartidos. En estos sistemas, un conjunto de transportistas forma una coalición para entregar coordinadamente las mercancías a los clientes finales. Los clientes compartidos son aquellos que pueden ser atendidos por más de un transportista de la coalición. Algunos estudios previos en los que se trata y analiza este enfoque colaborativo se encuentran en Fernández et al. [2018], Padmanabhan et al. [2020] y Mrad et al. [2023].

En el presente trabajo se propone una Búsqueda Tabú Multi-entorno para el *Shared Customer Collaboration Vehicle Routing Problem* (SCC-VRP) Fernández et al. [2018]. El problema considera la existencia de un conjunto de clientes que demandan mercancía de un conjunto de transportistas. Los clientes pueden agruparse en dos categorías atendiendo a si demandan mercancía de un único transportista o, por el contrario, esperan mercancía de más de uno de ellos. La demanda de los clientes puede ser atendida por cualquiera de los transportistas que lo tienen como cliente. Existe, por tanto, la posibilidad de que un cliente particular reciba la mercancía demandada a un transportista desde otro, siempre que también demande mercancía a este último. Los clientes que demandan mercancía a un único transportista deben ser, obligatoriamente, atendidos por este. En el problema se debe determinar la asignación de clientes a transportistas y las rutas que deben seguir los vehículos de reparto para minimizar los costes de transporte.

El resto de este artículo está organizado de la siguiente manera. En la Sección 2 se revisa la bibliografía relacionada con el problema abordado en este trabajo. La Sección 3 describe el problema de optimización, estableciendo cuáles son el objetivo y las restricciones que se deben considerar. La Sección 4 presenta el algoritmo de Búsqueda Tabú Multi-entorno diseñado en este trabajo para resolver el problema. La Sección 5 muestra los resultados computacionales alcanzados con la Búsqueda Tabú, así como la comparativa con los mejores resultados de la literatura científica asociada. Por último, la Sección 6 presenta las conclusiones y los trabajos futuros propuestos.

## 2.  Revisión bibliográfica

El modelo propuesto en Fernández et al. [2018] ha sido, hasta el momento, poco estudiado en la literatura científica. Hasta donde sabemos, el único trabajo posterior que aborda este problema es el de Torres-Ramos et al. [2019]. En él se propone un algoritmo basado en GRASP con búsqueda local iterativa (*Iterated Local Search*, ILS). Los autores generan un nuevo conjunto de instancias y realizan un análisis comparativo entre las soluciones heurísticas y las óptimas. Demuestran la eficacia y la eficiencia de la metaheurística, en comparación con los modelos matemáticos.

En Rekik et al. [2018] se considera un VRP con ventanas de tiempo flexibles dentro de un esquema cooperativo horizontal, en el que un conjunto de transportistas decide crear un sistema de distribución conjunto compartiendo depósitos y clientes. Deben diseñar y planificar las rutas de reparto con el objetivo de minimizar los costes de viaje y el número de violaciones de las ventanas de tiempo. Para este problema se proponen procedimientos heurísticos híbridos.

En Paul et al. [2019] se introduce el *Shared Capacity Routing Problem* (SC-VRP), un problema de diseño de rutas con capacidad compartida. En un SC-VRP existe un depósito central y puntos de transferencia desde los que se sirve la demanda de los clientes. Un cliente se dice exclusivo si solo debe ser visitado en una ruta que parte y finaliza en el depósito central. Si, por el contrario, debe ser visitado por rutas que parten y finalizan en el depósito central y en un punto de transferencia se llama cliente compartido. Los clientes exclusivos solo demandan al depósito central, mientras que los clientes compartidos demandan al depósito central y a algún punto de transferencia. Para evitar, en la medida de lo posible, que un cliente compartido sea visitado por dos vehículos, puede transferirse su demanda al depósito central hasta el punto de transferencia correspondiente. De esta forma, será solo un vehículo el que lo visite, entregando la mercancía demandada al depósito central y al punto de transferencia. Los autores proponen un procedimiento exacto y un algoritmo heurístico para resolver este problema. A diferencia del SC-VRP, en el *Shared Customer Collaboration Vehicle Routing Problem* no se considera la transferencia de mercancías entre los depósitos de los transportistas.

## 3.  Descripción del problema

El problema colaborativo de rutas de vehículos con clientes compartidos (*Shared Customer Collaboration Vehicle Routing Problem*, SCC-VRP) es un problema de optimización combinatoria NP-difícil en el que diferentes compañías de transporte (transportistas) realizan operaciones logísticas de reparto en una misma área urbana. A diferencia de lo que sucede en los problemas clásicos de rutas de vehículos, en los que las compañías consideran únicamente a sus clientes asignados, en el SCC-VRP, algunos clientes pueden ser atendidos indistintamente por cualquiera de los transportistas a los que demanden mercancía. En este contexto, los transportistas pueden transferir clientes compartidos entre ellos cuando se genere una disminución en el coste de distribución global. De esta manera, los clientes se asignan a la ruta del transportista que mejore la eficiencia de la entrega de la última milla, con el objetivo de minimizar los costes de transporte.

Sean $C$ un conjunto de $m$ transportistas independientes, cada una de ellas operando desde un depósito con su propia flota homogénea de vehículos de capacidad $Q$, y $N$ un conjunto de $n$ localizaciones de reparto, donde algunos clientes demandan únicamente a un transportista, mientras que otros lo hacen a varios de ellos. Sea $D$ el conjunto de depósitos independientes, con $o_r$ el depósito asignado al transportista $r \in C$. Sea $G = (V, A)$ un grafo dirigido completo, donde $V = N \cup (\cup_{r \in C} \{o_r\})$ está compuesto por el conjunto de todos los clientes más los depósitos de todos los transportistas, y $A = V \times V$ es el conjunto total de arcos que conectan los clientes y los depósitos.

Cada cliente $i \in N$ dispone de una demanda $d_i^r \geq 0$ asociada al transportista $r$. Si $d_i^r > 0$, entonces $i$ se considera un cliente de $r$. Sea $N_r$ el conjunto de clientes del transportista $r \in C$. Asimismo, sea $C_i$ el conjunto de transportistas que reciben demanda del cliente $i \in N$. El mecanismo de colaboración propuesto en el SCC-VRP implica que el subconjunto de clientes compartidos, que tienen requerimiento de demanda para distintos transportistas, $|C_i| > 1$, puede ser atendido por diferentes operadores, transfiriendo la demanda para optimizar los costos totales de distribución. En este caso, la demanda $d_i^s$ del cliente $i$ para el transportista $s$ puede ser transferida a cualquier transportista $r \in C_i$. El resto de los clientes, que demandan a un único transportista, $|C_i| = 1$, deben ser asignados a alguna de las rutas de dicho transportista. La demanda de un cliente compartido puede ser satisfecha por diferentes transportistas, pero cada parte de la demanda debe asignarse

completamente a uno solo de ellos; es decir, la demanda realizada a un transportista no puede ser dividida. Note que en el SCC-VRP, un transportista $r \in C$ visita a un subconjunto de $N_r$, por lo que en sus rutas asociadas, los únicos arcos permitidos son los pertenecientes al conjunto $A^r = \{(i, j) \in A : i, j \in N_r, (i = o_r \text{ y } j \in N_r), \text{o } (i \in N_r \text{ y } j = o_r)\}$.

El objetivo del SCC-VRP es minimizar el coste total de transporte, considerando la distribución de la demanda de los clientes compartidos como un factor para reducir dicho coste. La función objetivo se establece de la siguiente manera:

$$\min \sum_{r \in C} \sum_{(i,j) \in A_r} c_{ij} x_{ij}^r, \tag{1}$$

donde $c_{ij}$ representa el coste de transporte entre las localizaciones $i$ y $j$, y $x_{ij}^r$ es una variable de decisión binaria que indica si el transportista $r$ usa el arco $(i, j)$ en sus rutas. Se refiere al lector interesado al artículo de Fernández et al. [2018] para la revisión de las diferentes formulaciones matemáticas propuestas en la literatura científica, y que no son objeto de estudio en este trabajo. En ellas, el conjunto de restricciones garantiza el cumplimiento de todas las particularidades asociadas al SCC-VRP.

La Figura 1 muestra un ejemplo ilustrativo del problema de rutas de vehículos abordado en este trabajo. En este caso, los clientes 4 y 9 demandan mercancía a los dos transportistas, cuyos depósitos se ubican en las localizaciones $A$ y $B$, respectivamente, por lo que pueden ser atendidos por cualquiera de ellos. El resto de los clientes están asociados únicamente a uno de los transportistas, lo que se indica por los colores azul y amarillo. La subfigura (a) muestra una solución al problema clásico de rutas de vehículos con múltiples depósitos, en el que los clientes no están compartidos entre diferentes transportistas. En esta configuración, se observa que los clientes 4 y 9, que demandan mercancía a los dos transportistas, aparecen tanto en la ruta azul como en la ruta amarilla. Por otro lado, en la subfigura (b) se observa que la mercancía que demanda el cliente 9 a ambos transportistas es servida únicamente por la ruta amarilla. Lo mismo sucede con el cliente 4, que es enteramente atendido por la ruta azul. De esta manera, se genera una reducción significativa en los costes operativos de transporte, optimizando la logística de la distribución de la última milla que persigue el SCC-VRP.

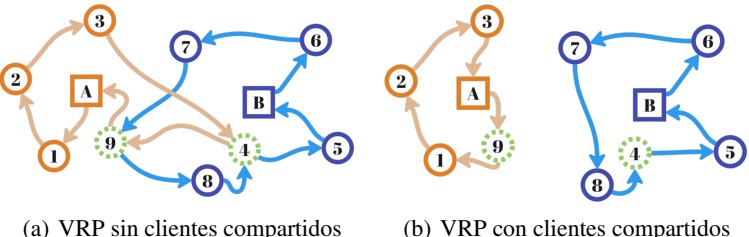

(a) VRP sin clientes compartidos          (b) VRP con clientes compartidos

Figura 1: Soluciones del SCC-VRP

El SCC-VRP es un problema NP-difícil, dado que generaliza el Problema de Rutas de Vehículos con Múltiples Depósitos (MDVRP) y el Problema de Rutas de Vehículos de Entrega Dividida (SDVRP), ambos conocidos por ser NP-difíciles.

## 4. Propuesta algorítmica

En esta sección, se presenta un algoritmo metaheurístico basado en Búsqueda Tabú Multi-Entorno y con Oscilación Aleatoria, diseñado e implementado en este trabajo para abordar el problema SCC-VRP. Por un lado, se considera el enfoque multi-entorno de la Búsqueda Tabú Wu et al. [2012], que permite explorar eficazmente el espacio de búsqueda mediante la combinación de las diferentes estructuras implementadas. Los multi-entornos se basan en la unión de los entornos subyacentes, en lugar de en la exploración secuencial convencional de los entornos básicos. Por otro lado, en el contexto de la Búsqueda Tabú (*Tabu Search*, TS), un algoritmo de Oscilación Estratégica Glover [1977], Glover et al. [1984] permite realizar la exploración del espacio de soluciones mediante la selección de un operador de búsqueda local, que varía según la fase que se deba ejecutar, ya sea

de intensificación o de diversificación. En el algoritmo presentado en este trabajo se hace uso de un procedimiento de oscilación aleatoria, mediante el cual en cada paso de la búsqueda local con múltiples entornos, se escoge uno de los entornos activos de forma aleatoria. La elección inteligente guiada por mecanismos de memoria a medio y largo plazo propia de la oscilación estratégica se propone como trabajo futuro. Sin embargo, el algoritmo propuesto garantiza el correcto balance entre las fases de intensificación y diversificación mediante el uso de estructuras de memoria a corto plazo y de la oscilación aleatoria.

La Búsqueda Tabú propuesta está diseñada para mejorar iterativamente una solución factible obtenida a partir de heurísticas constructivas. Como se muestra en el pseudocódigo presentado en el Algoritmo 1, el proceso descrito a continuación se repite durante un número específico de iteraciones (línea 9). Cada una de ellas se inicia con la generación de una solución candidata mediante una versión aleatorizada de la heurística *Clarke-Wright Savings* (CWS) (línea 10) Caceres-Cruz et al. [2014], Torres-Ramos et al. [2019]. A partir de esta solución inicial, el algoritmo aplica una estrategia de mejora que combina fases de perturbación aleatoria y oscilación para optimizar la solución actual, seguida de una fase de búsqueda tabú. El algoritmo se estructura en las siguientes etapas generales:

1. **Inicialización:** Se genera una solución inicial $S'$ mediante la heurística CWS aleatorizada (línea 10).

2. **Fase de mejora (diversificación-intensificación):**

   - Se ejecuta un procedimiento de búsqueda local (líneas 11-16) en el que, en cada iteración, se perturba ligeramente una copia de $S'$ (denotada por $S''$), utilizando un movimiento de reinserción aleatorio (línea 13), se realiza un único movimiento aleatorio a cualquiera de las rutas de la solución.

   - A continuación, se ejecuta una búsqueda local con oscilación aleatoria entre los distintos entornos diseñados para el SCC-VRP (línea 14). La oscilación aleatoria selecciona aleatoriamente, en cada paso de la búsqueda local, un operador de búsqueda local activo hasta que la solución actual no pueda ser mejorada por ninguno de ellos, alcanzando así un óptimo local con respecto a todos los entornos.

     Los movimientos empleados en este trabajo corresponden a operadores clásicos del VRP, incluyendo la reinserción y el intercambio, tanto dentro de la misma ruta como entre rutas diferentes asignadas al mismo transportista o a otro que comparta los clientes a reubicar. Además, se utiliza el operador 2-opt para la eliminación de cruces, disminuyendo el coste de transporte de las rutas generadas.

     Se destaca que en el problema SCC-VRP, las rutas pueden pertenecer a diferentes transportistas, por lo que los movimientos de reinserción e intercambio de clientes pueden afectar a múltiples transportistas, siempre que los clientes sean compartidos entre ellos. La evaluación eficiente de estos movimientos permite ampliar la capacidad exploratoria del algoritmo sin incrementar significativamente el tiempo de cómputo.

   - Si $S''$ mejora a $S'$, entonces $S'$ se actualiza a $S''$ (líneas 15-16).

3. **Actualización de la mejor solución:** Si la solución $S'$ produce un coste total inferior al de la mejor solución encontrada hasta el momento durante el proceso de búsqueda ($S^*$), esta se reemplaza por la anterior (líneas 17-18).

4. **Búsqueda Tabú:**

   - La fase de Búsqueda Tabú se realiza durante un número de iteraciones, *Tabu iterations* (línea 20). En cada una de ellas, a partir de la solución actual, $S'$, se genera un multi-entorno, formado utilizando todos los operadores de movimiento implementados (reinserción, intercambio intra y entre rutas y 2-opt) y se busca la mejor solución del multi-entorno que no sea tabú o que, siéndolo, cumpla el criterio de aspiración por objetivo global de la Búsqueda Tabú (líneas 21-25). Para ello, el algoritmo itera sobre un conjunto de movimientos candidatos, ordenados de menor a mayor coste total, y selecciona el mejor movimiento que no sea tabú o que, siendo tabú, cumpla con el criterio de aspiración por objetivo global de la Búsqueda Tabú (su mejora de coste es suficiente para superar la mejor solución conocida, $S^*$).

   - Si esa solución existe, se selecciona y se actualiza la memoria tabú (líneas 26-29). Cada demanda de cliente involucrada en el movimiento se marca como tabú durante un número de iteraciones especificado por el parámetro $tenure$. La lista tabú se implementa utilizando un mapa desordenado donde cada clave corresponde a una demanda y su valor asociado indica la

iteración hasta la cual esa demanda permanece tabú. Este diseño elimina la necesidad de borrar explícitamente las entradas de la lista tabú. Así, durante la evaluación de los movimientos candidatos, el algoritmo solo verifica si la iteración actual supera el valor de expiración tabú almacenado para cada demanda. Este enfoque minimiza la sobrecarga y mejora la eficiencia global del algoritmo.

- A continuación, se actualiza la mejor solución global de la búsqueda, $S^*$ (líneas 30-31) y se pasa a la siguiente iteración de Búsqueda Tabú.

5. **Finalización:** Finalmente, el procedimiento basado en Búsqueda Tabú Multi-Entorno y con Oscilación Aleatoria devuelve la mejor solución encontrada ($S^*$) a lo largo de todo el proceso de búsqueda en el espacio de soluciones.

## 4.1. Parámetros

El rendimiento del Algoritmo 1 depende críticamente de varios parámetros, que se ajustan cuidadosamente mediante el paquete $irace$ López-Ibáñez et al. [2016]. Los mejores valores obtenidos se describen a continuación:

- **Iteraciones de la Búsqueda Tabú** (línea 20): Se ejecutan 400 iteraciones para evaluar y aplicar movimientos, dentro del rango evaluado de 50 a 2000.

- **Período tabú (**$tenure$**)** (línea 29): Cada demanda afectada por un movimiento permanece tabú durante 10 iteraciones, con un rango explorado de 5 a 30.

Además, otros parámetros han sido configurados considerando un equilibrio entre eficiencia computacional y calidad de las soluciones obtenidas. Los valores empleados se describen a continuación:

- **Iteraciones globales del algoritmo**(*Algorithm iteration*) (línea 9), que incluye la fase inicial de generación aleatoria y mejora de la solución $S'$, así como la fase de Búsqueda Tabú, se ha fijado en 400 iteraciones.

- **Iteraciones de mejora de la búsqueda local** (línea 11): Se realizan 10 de iteraciones de perturbación y búsqueda local para mejorar la solución candidata.

- **Evaluación del movimiento candidato** (línea 21): Se mantienen 30 movimientos en una estructura de conjuntos múltiples ordenados por un comparador personalizado. Se priorizan los mejores movimientos y se permiten movimientos tabú si satisfacen el criterio de aspiración por objetivo global.

## 5. Experimentos computacionales

Esta sección describe los experimentos computacionales realizados para comparar el algoritmo de Búsqueda Tabú diseñado en este trabajo con los algoritmos que son estado del arte de la literatura para el problema SCC-VRP. En primer lugar, se considera como referencia el trabajo de Fernández et al. [2018], en el que se propone el SCC-VRP como una variante del problema de rutas de vehículos con múltiples depósitos (MDVRP) Cordeau et al. [1997]. Asimismo, se considera el trabajo de Torres-Ramos et al. [2019], que hasta donde se ha podido constatar, es el único artículo en la literatura que aborda el SCC-VRP mediante el uso de algoritmos metaheurísticos. Las pruebas se realizaron en un ordenador con procesador Intel Core i7-13620H de $13^a$ generación a 2,90 GHz y con 16 GB de RAM. El algoritmo propuesto fue implementado en C++ utilizando la versión 14 del estándar.

Los experimentos se realizan sobre tres conjuntos de instancias. En primer lugar, se consideran los conjuntos de instancias $S1$ y $S2$, generados en el trabajo de Fernández et al. [2018]. El conjunto $S1$ está compuesto por 12 instancias derivadas de las instancias propuestas por Cordeau et al. [1997] para el MDVRP. Estas instancias tienen entre 18 y 30 clientes, todas ellas con 2 transportistas, que comparten un subconjunto de los clientes, los cuales fueron clasificados como compartidos con probabilidad de $0,25$. El conjunto $S2$ está compuesto, a su vez, por dos subconjuntos, $S2_R$ y $S2_C$, cada uno de ellos con 50 instancias. En el subconjunto $S2_R$, los clientes se localizan de forma aleatoria en un cuadrado de 100 unidades de lado, mientras que en $S2_C$ se agrupan en clústeres dentro del mismo dominio. Ambos subconjuntos incluyen 2 transportistas, cuyos depósitos se localizan en las posiciones $(0,0)$ y $(100,100)$ del cuadrado generado. En cada subconjunto hay 10 instancias con

**Algoritmo 1** Multi-neighborhood Tabu Search with Random Oscillation

```
 1: procedure SOLVE USING TABÚ SEARCH
 2:     Parámetros:
 3:         algorithmIterations ← 400                                    ▷ Número de iteraciones de Globales
 4:         tabuIterations ← 200                                         ▷ Número de iteraciones de Búsqueda Tabú
 5:         localSearchIterations ← 10                                   ▷ Iteraciones de mejora local
 6:         tenure ← 10                                                  ▷ Período tabú
 7:         bestMovementsTableSize ← 30                                  ▷ Tamaño de la tabla de mejores movimientos
 8:     cost(S*) ← +∞                                                    ▷ Mejor Solución
 9:     for each algorithmIterations do
10:         S' ← GenerateNewSolutionUsingRandomizedCWS()
11:         for each localSearchIterations do
12:             S'' ← S'
13:             S'' ← perturbate(S'')
14:             S'' ← search(S'')
15:             if cost(S'') < cost(S') then
16:                 S' ← S''
17:         if cost(S') < cost(S*) then
18:             S* ← S'
19:         initialize(TabuList)
20:         for each tabuIterations do
21:             bestMovements ← GetBestMovesFromAllMovements(S', bestMovementsTableSize)
22:             for each nextMove ∈ sorted(bestMoves − worstMoves) do
23:                 if IsNotTabu(relatedDemands(nextMove)) or cost(nextMove) < cost(S*) then
24:                     selectedMove ← nextMove
25:                     break;
26:             if ExistsPossibleMove then
27:                 S' ← ApplyMove(S', selectedMove)
28:                 for each demand ∈ demands(selectedMove) do
29:                     tabuList[demand] ← actualIterations + tenure
30:                 if cost(S') < cost(S*) then
31:                     S* ← S'
32:     return S*
```

los números de clientes $|N| = \{10, 15, 20, 25, 30\}$. Al igual que en las instancias del conjunto $S1$, cada cliente tiene una probabilidad de $0{,}25$ de ser compartido entre los dos transportistas.

Por último, el tercer conjunto de instancias considerado, $S3$, ha sido generado en este trabajo siguiendo los criterios establecidos por Torres-Ramos et al. [2019] para la generación de sus propias instancias, ya que estas no están disponibles. El conjunto $S3$ está formado por 12 nuevas instancias con un número de clientes es más elevado, $|N| = \{90, 180, 270, 360\}$. Se han generado 3 instancias para cada número de clientes, cada una de ellas con capacidades de los vehículos de 100, 250 y 400, respectivamente. Todas las demandas se escogen de forma uniforme en el rango $(5, 25)$. El conjunto de clientes compartidos representa un $20\,\%$ del total. Por último, se consideran también 2 transportistas en todas las instancias generadas.

Los resultados obtenidos en estos conjuntos de instancias utilizando el algoritmo de Búsqueda Tabú Multi-entorno propuesto en este trabajo han sido comparados con los mejores resultados conocidos alcanzados por el algoritmo GRASP $\times$ ILS desarrollado en Torres-Ramos et al. [2019]. Hasta donde sabemos, es el único algoritmo metaheurístico propuesto en la literatura para abordar el SCC-VRP. El algoritmo GRASP $\times$ ILS genera una solución inicial con GRASP mediante el algoritmo *Clarke-Wright Savings* (CWS) Caceres-Cruz et al. [2014], Torres-Ramos et al. [2019], combinando aleatorización y búsqueda local para la intensificación. A continuación, aplica una serie de perturbaciones controladas en ILS para ejecutar la fase de diversificación de la búsqueda, seguidas de un procedimiento de búsqueda local.

En el trabajo de Torres-Ramos et al. [2019] se muestran resultados para el conjunto de instancias $S1$ generado por Fernández et al. [2018], pero no para el conjunto $S2$. Además, tampoco ha sido posible comparar con las instancias generadas en dicho trabajo, ya que sus autores no las tenían disponibles. Por ello, para realizar la comparativa de algoritmos sobre los tres conjuntos considerados, hemos replicado la implementación del algoritmo GRASP $\times$ ILS. Por un lado, se ha implementado dicho algoritmo, manteniendo sus parámetros, pero optimizando su eficiencia, logrando reducciones de

| Instances | GRASPxILS replicado | | GRASPxILS mejorado | | GRASPxILS | | Búsqueda Tabú | | |
|---|---|---|---|---|---|---|---|---|---|
| **S1** | **Coste** | **t(s)** | **Coste** | **t(s)** | **Coste** | **t(s)** | **Coste** | **t(s)** | **%GAP** |
| **vrps_1L** | 273,77 | 9,86 | 273,77 | 0,76 | 273,77 | 2,92 | 273,77 | 2,16 | 0,00 |
| **vrps_2L** | 317,51 | 12,92 | 317,51 | 1,13 | 317,51 | 4,14 | 317,42 | 2,96 | -0,02 |
| **vrps_3L** | 233,28 | 4,45 | 233,28 | 0,62 | 233,28 | 2,52 | 233,28 | 1,59 | 0,00 |
| **vrps_4L** | 322,30 | 10,76 | 322,30 | 0,94 | 322,30 | 4,36 | 322,30 | 2,70 | 0,00 |
| **vrps_5L** | 320,28 | 11,53 | 320,28 | 0,94 | 320,28 | 4,07 | 320,28 | 2,64 | 0,00 |
| **vrps_6L** | * 230,08 | 5,61 | 230,08 | 0,88 | 230,08 | 3,69 | * 230,08 | 2,16 | 0,00 |
| **vrps_7L** | * 156,93 | 6,66 | 157,36 | 0,31 | 157,36 | 1,19 | 157,36 | 1,12 | 0,27 |
| **vrps_8L** | 239,05 | 11,04 | 239,05 | 1,30 | 239,05 | 5,71 | * **237,83** | 3,40 | -0,51 |
| **vrps_9L** | * 392,06 | 3,63 | 392,06 | 0,48 | 392,06 | 2,20 | * 392,06 | 1,41 | 0,00 |
| **vrps_10L** | * 455,71 | 5,56 | 455,71 | 0,41 | 455,71 | 1,86 | * 455,71 | 1,32 | 0,00 |
| **vrps_11L** | * 486,90 | 4,47 | 486,90 | 0,30 | 486,90 | 1,25 | * 486,90 | 0,94 | 0,00 |
| **vrps_12L** | 749,81 | 7,52 | 749,81 | 1,31 | 749,81 | 5,18 | 749,81 | 2,73 | 0,00 |

Tabla 1: Resultados para las instancias del conjunto $S1$ (Fernández et al. [2018]) (* Óptimo)

tiempo de hasta un $95\%$ en una máquina equivalente a la de Torres-Ramos et al. [2019]. Por otro lado, se han ajustado correctamente los parámetros de este algoritmo replicado, generando el algoritmo al que denominamos GRASP $\times$ ILS mejorado, que alcanza mejoras significativas en los costes de transporte para los conjuntos de instancias $S2$ y $S3$.

La Tabla 1 muestra la comparativa de resultados para el conjunto de instancias $S1$. La primera columna indica el identificador de la instancia. Las siguientes dos columnas contienen los valores de coste y tiempo del algoritmo GRASP $\times$ ILS presentados en el trabajo de Torres-Ramos et al. [2019] para este conjunto de instancias. Las siguientes cuatro columnas muestran el coste mínimo y el tiempo obtenidos por los algoritmos GRASP $\times$ ILS replicado y GRASP $\times$ ILS mejorado implementados por los autores del presente trabajo. Posteriormente, se proporciona la misma información para el algoritmo de Búsqueda Tabú propuesto. Finalmente, la última columna muestra los valores de GAP para el coste mínimo de la Búsqueda Tabú, comparado con los resultados del algoritmo GRASP $\times$ ILS reportados en el trabajo de sus autores. Los valores representados en negrita muestran qué soluciones son las óptimas para esas instancias tal como se muestra en el trabajo de Fernández et al. [2018].

En este tabla se observa, en primer lugar, que el algoritmo GRASP $\times$ ILS replicado obtiene los mismos resultados que los mostrados artículo de Torres-Ramos et al. [2019] para su implementación del algoritmo, salvo para la instancia $vrps_{7L}$, en la que obtiene una solución de peor calidad, con un GAP de $0,27\%$. Como se ha indicado anteriormente, nuestro algoritmo replicado es hasta un $95\%$ más eficiente, por lo que mostramos también los resultados alcanzados con un nuevo ajuste de parámetros para este algoritmo. Se observa que el algoritmo GRASP $\times$ ILS mejorado alcanza los mismos resultados que el GRASP $\times$ ILS replicado para este conjunto formado por 12 instancias. Se aprecian mayores diferencias para los otros conjuntos de instancias usados en este trabajo, tal como se mostrará en las restantes tablas de resultados.

Por otro lado, se observa que la Búsqueda Tabú mejora los mejores resultados conocidos para las instancias $vrps\_2L$ y $vrps\_8L$. En el caso de esta última instancia, se alcanza la solución óptima presentada en el trabajo de Fernández et al. [2018]. Asimismo, obtiene los mejores resultados conocidos para 9 instancias y solo empeora en la instancia $vrps\_7L$, en la que se obtiene un GAP del $0,27\%$.

La Tabla 2 muestra la comparativa de resultados entre el algoritmo de Búsqueda Tabú, el GRASP $\times$ ILS replicado y el GRASP $\times$ ILS mejorado para los subconjuntos de instancias $S2_R$ y $S2_C$ en el lado izquierdo y derecho de la tabla, respectivamente. Cada fila de la tabla representa el promedio de los resultados para las 10 instancias correspondientes a cada número de clientes $|N| = \{10, 15, 20, 25, 30\}$ de los subconjuntos $S2_R$ y $S2_C$. En en lado izquierdo de la tabla, la primera columna muestra el número de clientes de las instancias. Las siguientes dos columnas muestran el coste y el tiempo computacional obtenidos por el GRASP $\times$ ILS replicado. La misma información se representa para el GRASP $\times$ ILS mejorado y la Búsqueda Tabú en las siguientes cuatro columnas. En el lado derecho de la tabla se presenta la misma información para las instancias del subconjunto $S2_C$. Se ha marcado en negrita el mejor valor promedio obtenido. Se observa que para las instancias del conjunto $S2_R$, los algoritmos GRASP $\times$ ILS mejorado y Búsqueda Tabú alcanzan los mejores valores promedio, obteniendo los mejores resultados en 3 de los 5 subconjuntos de 10 instancias.

Sin embargo, para las instancias del conjunto $S2_C$, la Búsqueda Tabú alcanza 4 de los 5 mejores promedios, mostrándose más eficaz que los algoritmos GRASP $\times$ ILS.

| \|N\| | GRASPxILS replicado | | GRASPxILS mejorado | | Búsqueda Tabú | | \|N\| | GRASPxILS replicado | | GRASPxILS mejorado | | Búsqueda Tabú | |
|---|---|---|---|---|---|---|---|---|---|---|---|---|---|
| **S2_R** | Coste | t(s) | Coste | t(s) | Coste | t(s) | **S2_C** | Coste | t(s) | Coste | t(s) | Coste | t(s) |
| 10 | 522,55 | 0,03 | 522,55 | 0,32 | **521,88** | 0,66 | 10 | **432,87** | 0,02 | 432,96 | 0,30 | **432,87** | 0,57 |
| 15 | 624,82 | 0,05 | **622,71** | 0,57 | 624,65 | 1,27 | 15 | 482,50 | 0,04 | 482,50 | 0,54 | **482,49** | 1,25 |
| 20 | **713,64** | 0,09 | **713,64** | 1,09 | **713,64** | 2,41 | 20 | 562,26 | 0,10 | 561,81 | 1,24 | **561,39** | 2,33 |
| 25 | 873,67 | 0,15 | 872,59 | 1,81 | **868,47** | 3,28 | 25 | 693,60 | 0,19 | 691,62 | 2,21 | **691,36** | 3,54 |
| 30 | 961,32 | 0,28 | **958,21** | 3,36 | 960,23 | 4,95 | 30 | 775,16 | 0,35 | **767,44** | 4,16 | 768,46 | 5,31 |

Tabla 2: Resultados promedio para las instancias del conjunto $S2$ (Fernández et al. [2018])

El análisis de los resultados sobre el conjunto de instancias $S3$ (Tabla 3) muestra que la Búsqueda Tabú Multi-entorno supera a las variantes de GRASP $\times$ ILS, empleando una asignación adecuada para instancias más grandes. Para optimizar el rendimiento, se ajustaron los parámetros clave del algoritmo, se establecieron 100 iteraciones globales, 50 iteraciones de Búsqueda Tabú y manteniendo un tenure de 10.

Dado el gran espacio de soluciones, se priorizó una mejor solución inicial con 100 iteraciones de mejora en la búsqueda local, además se fijó la tabla de mejores movimientos en 5, reduciendo el costo computacional. Como resultado, la Búsqueda Tabú obtuvo 11 mejores soluciones frente a las 1 de la variante de GRASP $\times$ ILS mejorada en este trabajo. Además, la Búsqueda Tabú destaca en instancias con mayor número de clientes, especialmente en aquellas con 180, 270 y 360 clientes. Por último, la Tabla 4 resume los resultados, indicando cuántas de las mejores soluciones conocidas alcanza cada algoritmo en el total de instancias de los conjuntos $S1$, $S2$ y $S3$.

| Instances | GRASPxILS replicado | | GRASPxILS mejorado | | Búsqueda Tabú | | |
|---|---|---|---|---|---|---|---|
| **S3** | Coste | t(s) | Coste | t(s) | Coste | t(s) | %GAP |
| **scc_vrp_0000** | 1.426,94 | 1,31 | 1.387,92 | 2,18 | **1337,91** | 2,15 | -3,74 |
| **scc_vrp_0001** | 822,77 | 1,46 | **758,39** | 2,45 | 780,13 | 2,68 | 2,79 |
| **scc_vrp_0002** | 587,80 | 1,82 | 558,69 | 3,03 | **552,44** | 3,49 | -1,23 |
| **scc_vrp_0003** | 2.587,69 | 6,36 | 2.485,24 | 9,84 | **2476,74** | 9,23 | -0,34 |
| **scc_vrp_0004** | 1.589,52 | 7,62 | 1.453,58 | 12,17 | **1395,61** | 11,53 | -4,15 |
| **scc_vrp_0005** | 1.168,43 | 8,70 | 1.141,96 | 14,42 | **1118,04** | 14,37 | -2,14 |
| **scc_vrp_0006** | 3.936,03 | 20,87 | 3.808,26 | 29,78 | **3685,68** | 27,21 | -3,33 |
| **scc_vrp_0007** | 2.157,73 | 23,30 | 2.087,12 | 33,89 | **1931,20** | 30,03 | -8,07 |
| **scc_vrp_0008** | 1.895,96 | 25,06 | 1.727,07 | 36,82 | **1684,21** | 31,64 | -2,54 |
| **scc_vrp_0009** | 5.026,45 | 46,80 | 4.779,27 | 65,59 | **4683,20** | 53,24 | -2,05 |
| **scc_vrp_0010** | 2.905,40 | 50,99 | 2.777,44 | 70,84 | **2637,23** | 58,78 | -5,32 |
| **scc_vrp_0011** | 2.364,66 | 52,57 | 2.231,99 | 84,17 | **2076,52** | 67,74 | -7,49 |

Tabla 3: Resultados para las instancias del conjunto $S3$

Los resultados computacionales demuestran que el algoritmo basado en Búsqueda Tabú propuesto es altamente competitivo para el SCC-VRP. Se observa una mejora significativa en la calidad de las soluciones, con reducciones notables en el coste total de distribución y tiempos de cómputo aceptables incluso en instancias grandes. Además, la metodología muestra gran adaptabilidad a distintos escenarios, validando la hipótesis de que la integración de estrategias de colaboración entre transportistas con algoritmos de Búsqueda Tabú optimiza eficientemente las rutas de transporte en enfoques colaborativos.

| Set of Instances | GRASPxILS replicado | GRASPxILS mejorado | Búsqueda Tabú |
|---|---|---|---|
| $S1$ | 10 | 10 | 12 |
| $S2_R$ | 39 | 42 | 44 |
| $S2_C$ | 32 | 39 | 46 |
| $S3$ | 0 | 1 | 11 |

Tabla 4: Número de mejores soluciones obtenidas

## 6. Conclusiones y trabajo futuro

Este trabajo propone un algoritmo eficiente de Búsqueda Tabú Multi-entorno con oscilación aleatoria para el problema de rutas de vehículos con clientes compartidos (SCC-VRP). Los resultados experimentales demuestran que el enfoque propuesto genera soluciones de alta calidad, superando las obtenidas en la literatura previa. Además, se confirma la efectividad de la planificación colaborativa de rutas en comparación con los enfoques tradicionales, destacando su potencial para optimizar la distribución en escenarios con múltiples actores.

En cuanto a trabajos futuros, se plantea la ampliación del problema para integrar restricciones adicionales, como ventanas de tiempo y variabilidad en la disponibilidad de recursos. Asimismo, se explorará el uso de técnicas de aprendizaje automático para guiar la Búsqueda Tabú hacia regiones prometedoras del espacio de soluciones y para el ajuste dinámico de sus parámetros. Finalmente, se prevé extender los experimentos a instancias de gran escala y escenarios con más de dos depósitos, fortaleciendo la aplicabilidad del enfoque en contextos más complejos.

## Agradecimientos

Este trabajo ha sido financiado por la Agencia Estatal de Investigación (España; proyecto PID2019-104410RB-I00/AEI/10.13039/501100011033).

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
