# OpenReview forum: "Búsqueda Tabú Multi-Entorno para el Problema Colaborativo de Rutas de Vehículos con Clientes Compartidos"
_MAEB/2025/Congreso — MAEB 2025_

### Official Review · Reviewer_AUSe · 2025-03-11
**Revisión del Artículo: "Búsqueda Tabú Multi-Entorno para el Problema Colaborativo de Rutas de Vehículos con Clientes Compartidos"**

**Rating:** 4
**Confidence:** 4

**Review:**

El artículo presenta un enfoque metaheurístico basado en TS para resolver el problema colaborativo de rutas de vehículos con clientes compartidos (SCC-VRP). Este problema surge en la logística urbana y busca optimizar la distribución de la última milla, reduciendo costos operativos y mejorando la utilización de recursos. El artículo, en líneas generales, está bien estructurado y presenta una revisión bibliográfica adecuada. Los experimentos demuestran que el algoritmo propuesto es competitivo en términos de calidad de solución y tiempo de ejecución en comparación con los algoritmos existentes. Además, utilizan un programa de ajuste de parámetros para encontrar los valores óptimos de los parámetros tanto del algoritmo propuesto como de los algoritmos de comparación.

La investigación es de interés para la comunidad del MAEB y, por lo tanto, recomiendo su aceptación. Sin embargo, hay varios aspectos que requieren atención y que pueden mejorar considerablemente la calidad del artículo.

1.  Es condición indispensable para la aceptación en el congreso que los trabajos se ajusten estrictamente a la normativa. Los agradecimientos no deberían ser visibles en esta versión.

2. La Figura 1 tiene una leyenda poco descriptiva, por favor, añadid una descripción más detallada de la figura. Además, las imágenes tienen un sombreado o un fondo que se podría eliminar para mejorar la calidad de la impresión.

2. Los Cuadros 1, 2, 3 y 4 tienen encabezados en inglés: "Set of Instances", "Cost", "Time", etc. Por favor, revisad tanto las tablas como la descripción de las columnas para asegurar que estén en español.

3. Cuando se utilicen acrónimos cuyo origen viene de un término en inglés (ej., ILS) se recomienda añadir la traducción al español entre paréntesis la primera vez que se menciona (ej., Búsqueda Local Iterada (ILS, del inglés, \textit{Iterated Local Search})). El acrónimo VRP no ha sido definido. Otra opción es mantener el nombre original y mantenerlo en cursiva. Creo que la comunidad de investigadores del MAEB estará familizarizada con los algoritmos y no requieren de su traducción.

4. La descripción formal del problema no está clara. Por ejemplo, $C$ es un conjunto de $m$ compañías, entonces $|C|=m$. Por lo tanto, se presupone que $C$ contiene compañías, pero, en la línea 116 se indica que $r$ es un transportista y que $C$ se compone de transportistas ya que $r \in C$. ¿Es lo mismo un transportista que una compañía? ¿Las compañías solo pueden tener un único transportista? Por favor, revisad la descripción del problema para asegurar que sea clara y coherente.

5. En relación con el punto anterior, la definición del conjunto $V$ me ha parecido difícil de leer porque el índice del operador unión aparece a la derecha del símbolo en vez de debajo. Quizás esto se lea mejor: $\{o_r : r \in C\}$ en una línea. Esto es simplemente una recomendación, no hay nada incorrecto.

6. Respecto al Algoritmo 1, este diría que es lo que más atención requiere para mejorar el artículo. Está detallado principalmente desde el punto de la programación y puede resultar complejo para un público no especializado (en la orientación a objetos). Recomiendo rehacer  el algoritmo teniendo en cuenta las siguientes recomendaciones:
    - BS.cost podría ser simplemente BS y para acceder a su coste se llama a la función cost en la línea 30.
    - Para saber el coste de una S en las líneas 9 y 13 se hace mediante la ¿función/atributo? "cost" (S.cost) pero en la línea 30 se llama a una función (cost(BS)). Creo que esta inconsistencia puede llevar a confusión, creo que sería más claro si siempre lo hacéis como en la línea 30.
    - En el bucle for del paso 3 no está claro que es "Algorithm Iteration", hay que leer la siguiente sección para entenderlo. Sería más adecuado indicar algo más descriptivo for i = 0 hasta Imax, por ejemplo, y describir en el texto que es el número máximo de iteraciones y cómo se ha seleccionado. De manera similar se haría con los bucles de los pasos 5, 17, 20 y 27.
    - Pasos 7 y 8, ¿qué es "ro"?
    - Considerad escribir ciertos terminos o los comentarios en español.
    - El estilo de inicializar las variables del algoritmo no es consistente. Por ejemplo, en el paso 2 se asigna un valor y en el paso 16 se ejecuta una función.

7.  Respecto al ajuste de parámetros del algoritmo, se indica que el número de iteraciones globales es 400. ¿Tiene sentido que el número de iteraciones globales sea un parámetro a ajustar? ¿No debería ser un parámetro fijo que determine el investigador en función de las necesidades concretas (tiempo, calidad, convergencia, etc.)? Entiendo que si irace ajusta el número de iteraciones globales, cuantas más iteraciones se hagan, mejor será el resultado y por lo tanto, no tiene sentido ajustar este parámetro. Por favor, revisad este aspecto.

8. En la línea 257, incluiría una especificación del lenguaje de programación y la versión utilizada para implementar el algoritmo propuesto.

9. En la línea 303 se presenta la columna "GAP", definid qué es el GAP en este contexto ya que en distintos contextos puede significar cosas distintas. Por ejemplo, podría significar la separación entre el límite superior e inferior de la solución, o la desviación porcentual con respecto a una solución de referencia (como es el caso). En ese caso, puede ser más adecuado llamarlo "Desviación".

10. Denominar al GRASPxILS mejorado cuando lo único que tiene es un ajuste de parámetros es confuso e incita al lector a pensar que se le ha hecho alguna mejora al algoritmo ya sea en términos de calidad de solución o tiempo de ejecución. Recomiendo cambiar el nombre a "GRASPxILS ajustado" o algo similar.

11. Recomiendo incluir en las tablas una fila final con el agregado de los resultados para que el lector pueda comparar de forma rápida los resultados obtenidos.

12. En los Cuadros 2 y 3 las negritas representan mejores soluciones del experimento mientras que en el Cuadro 1 representan el óptimo global. Recomiendo que se utilice el mismo criterio en todos los cuadros para evitar confusiones.

13. En la línea 84 se indica que se propone un modelo matemático. En este artículo no se presenta ningún modelo matemático, por lo que recomiendo eliminar esta afirmación o simplemente indicar que se ha elaborado un modelo matemático que no se presenta en el artículo (totalmente entendidble dada la extensión máxima o guardar cierto contenido para una revista) pero que sí se ha utilizado para medir la calidad de las soluciones obtenidas. Aunque esto, tampoco está del todo claro, ya que, en la sección de resultados, en el cuadro 1 se presenta el óptimo global, pero no se indica cómo se ha obtenido, ¿se ha obtenido mediante el modelo matemático propuesto? Por favor, aclarad este punto.

14. De manera similar, en las conclusiones se mencionan un par de veces el modelo, pero no se ha presentado ningún modelo en el artículo. Simplemente, clarifcad este asunto.

15. Otros aspectos menores a mejorar, relacionados con la redacción y la presentación:

- Línea 2, evitaría la expresión "...representa un reto de optimización..." y la cambiaría simplemente por "...es un problema de optimización...".
- Línea 28-31, añadid una referencia, ya sea en la bibliografía o en nota a pie de página, para que el lector pueda corroborar esos datos.
- Línea 51: poner en formato cursiva "order sharing" y "capacity sharing".
- Los acrónimos GRASP e ILS no aparecen siempre con el formato adecuado, da la sensación que a veces están en entorno matemático.
- Líneas 155-157: revisad los acrónimos y las mayúsculas de las palabras, actualmente solo "Problema" tiene la primera letra en mayúscula.
- En el algoritmo 1, podéis hacer referencia a una línea en específica mediante "ref{paso1}" habiendo puesto "label{paso1}" en la línea que queréis referenciar.
- No hay consistencia en la forma de escribir "Búsqueda Tabú" a lo largo del artículo, a veces aparece en mayúsculas y otras en minúsculas.
- En el Cuadro 1, "S1" no aparece en mayúsculas. Hay algún paquete en latex que os permite escribir en entornos matemáticos letras y números en negrita (bm  o amsmath, por ejemplo).
- En el Cuadro 2, el número 713,64  de la 3a fila, 6a columna, tiene el último dígito sin negrita.
- No hay concistencia en la forma de escribir "GAP" a lo largo del artículo, a veces aparece en mayúsculas y otras en minúsculas.
- Línea 150: los dos transportista --> los dos transportistas
- Línea 181: una fase de busqueda tabú --> una fase de búsqueda tabú
- Línea 235: Número de iteracciones globales  --> Número de iteraciones globales
- Línea 308: los mostrados en el paper --> los mostrados en el artículo/estudio

---

### Official Review · Reviewer_rBiz · 2025-03-14
**Búsqueda Tabú Multi-Entorno para el Problema Colaborativo de Rutas de Vehículos con Clientes Compartidos**

**Rating:** 5
**Confidence:** 4

**Review:**

En este trabajo los autores proponen un enfoque metaheurístico basado en búsqueda tabú con exploración multi-entorno para resolver el problema colaborativo de rutas de vehículos con clientes compartidos (SCC-VRP, por sus siglas en inglés). El algoritmo propuesto presenta unos resultados que superan al algoritmo del estado del arte actual, respaldados por una experimentación robusta y que no deja lugar a dudas. Además, los autores re-implementan el algoritmo del estado del arte, mejorando su eficiencia en un porcentaje muy elevado, además de obtener un nuevo ajuste de parámetros que mejora las soluciones de la propuesta anterior. Aún con esta implementación mejorada, la nueva propuesta basada en búsqueda tabú y exploración multi-entorno proporciona mejores soluciones.

Cambios mayores:

No tengo comentarios mayores al respecto de este trabajo.

Typos y cambios menores:

- En la introducción, sería bueno añadir algunas referencias que apoyasen esta afirmación: "Según el Ministerio de Agricultura y Pesca, Alimentación y Medioambiente del Gobierno de España, el transporte de mercancías y personas es responsable del 25 % de las emisiones totales de gases de efecto invernadero, correspondiendo al transporte por carretera el 95 % de estas emisiones."

- Página 4, línea 150: los dos transportista --> los dos transportistas

- La notación S* se utiliza habitualmente para denotar la solución óptima. Sugiero modificar esta notación para que diga simplemente S.

- En el pseudocódigo del Algoritmo 1, sugiero sustituir ro.perturbate y ro.search por simplemente perturbate y search. Puede resultar extraño para lectores no familiarizados con la programación informática. En el mismo pseudocódigo, en la línea 21, la llamada a relatedDemands desde nextMove se ha compilado de forma extraña. Sucede lo mismo en la línea 27 con selectedMove.demands. En general, trataría de simplificar este pseudocódigo para alejarse del código fuente original. El propósito de un pseudocódigo es abstraer al lector del código fuente, simplificando la comprensión del algoritmo en sí. Considero que este pseudocódigo aún está muy cercano al código fuente.

- En la Sección 5, se afirma que el trabajo de Torres-Ramos et al. es el único trabajo en la literatura que enfoca el problema desde el punto de vista metaheurístico. Posteriormente, se afirma que es el último (líneas 280-281 de la página 7). Por favor, eliminar esta afirmación o listar trabajos previos que aborden el problema desde el punto de vista metaheurístico, o cambiar la afirmación para decir "primero" en lugar de "último", aunque en mi opinión la frasees redundante y debería eliminarse.

- En la Tabla 1, tratar de modificar el pie de la Tabla para que diga Tabla en lugar de Cuadro.

- También en la Tabla 1, sugiero seleccionar una marca diferente para los valores óptimos, y utilizar la negrita. para indicar los mejores valores obtenidos en el experimento, facilitando así el análisis de la tabla al permitir evaluar cuántas veces obtiene cada algoritmo la mejor solución para una instancia concreta del experimento, cuál lo hace en un menor tiempo y cuál obtiene un GAP menor, de un solo vistazo.

---

### Official Review · Reviewer_aZ9k · 2025-03-17
**Búsqueda tabú multi-entorno con oscilación aleatoria para el SCC-VRP**

**Rating:** 4
**Confidence:** 5

**Review:**

En este trabajo se propone un algoritmo que combina Búsqueda Tabú Multi-entorno con Oscilación Aleatoria para generar soluciones de alta calidad para el problema de rutas de vehículos con clientes compartidos (SCC-VRP). Los resultados obtenidos se comparan con los mejores resultados encontrados en la literatura, mostrando la superioridad de la propuesta.

## Cambios mayores

Se recomienda el uso de test estadísticos que respalden los resultados obtenidos. En muchos casos los resultados son similares y estos tests podrían validar la hipótesis del trabajo. Además, se puede observar que las instancias más pequeñas ya no son un reto para las propuestas algorítmicas actuales y no sirven para discriminar.

## Cambios menores

- l10: En todo el artículo, uniformizar la notación: los nombres de los algoritmos se escriben con mayúsculas en algunas partes y en minúsculas en otras
- l29: Incluir una referencia a ese dato
- l51: los términos en inglés deberían ir en cursiva
- l134: se indica que se "optimiza la distribución de la demanda de los clientes", pero según indica la ecuación 1 realmente solo se optimiza  el coste total de transporte, mientras que la distribución se utiliza para reducir ese coste. Si es así, reformular esa frase.
- l187: definir de forma más precisa la reinserción, ¿se hace una única reinserción? ¿Es en la misma ruta o en rutas diferentes?
- l219: el tiempo que un componente se marca como tabú se suele definir en la literatura como tenure, en lugar de timeOnTabu.
- l233: el uso de irace es correcto, pero es recomendable indicar qué valores se han utilizado durante las pruebas
- Algoritmo 1: especificar los parámetros de entrada en el pseudocódigo. El pseudocódigo es muy cercano al código real, se agredecería un poco más de abstracción.
- Algoritmo 1, l7-8: no está definido ro, ¿a qué se refiere?
- Algoritmo 1, l9: en general, en pseudocódigo es más recomendable usar funciones que una notación similar a la orientación a objetos, es decir S'.cost -> Cost(S'), por ejemplo

---

### Decision · Program_Chairs · 2025-03-20

Accept